# Different Configurations of Radio-Frequency Atomic Magnetometers—A Comparative Study

**DOI:** 10.3390/s22249741

**Published:** 2022-12-12

**Authors:** Patrick Bevington, Witold Chalupczak

**Affiliations:** The National Physical Laboratory, Hampton Road, Teddington, London TW11 0LW, UK

**Keywords:** atomic magnetometer, optical pumping, radio-frequency spectroscopy

## Abstract

We comprehensively explore different optical configurations of a radio-frequency atomic magnetometer in the context of sensor miniaturisation. Similarities and differences in operation principles of the magnetometer arrangements are discussed. Through analysis of the radio-frequency and noise spectra, we demonstrate that all configurations provide the same level of atomic polarisation and signal-to-noise ratio, but the optimum performance is achieved for significantly different laser powers and frequencies. We conclude with possible strategies for system miniaturisation.

## 1. Introduction

Detection systems based on radio-frequency (RF) atomic magnetometers have been successfully implemented in a wide range of measurement scenarios covering explosives detection [1], geomagnetic surveys [2,3], inductive measurements [4,5], and communication [6,7]. These radio-frequency magnetometers offer an improved sensitivity (fT Hz${}^{-1/2}$ range) [8,9] over standard pick-up coils combined with simpler instrumentation that enables miniaturisation, differentiating them from superconducting interference devices (SQUIDs). Some of the miniaturisation limits and the generic SWaP-C (size, weight, power and cost) requirements for a practical sensing device are defined by the laser system used in the operation of the magnetometer.

The principle of operation of rf atomic magnetometers relies on the detection of magnetically driven atomic coherences between neighbouring Zeeman sublevels of a given ground state level in the alkali–metal atoms. The atomic coherence is generated by an rf field resonant with the energy splitting of adjacent Zeeman sublevels. The sublevels are tuned into resonance with the rf field through the Zeeman effect, i.e., via the application of a bias magnetic field. Precessing atomic coherence is detected through its mapping onto the polarisation of a linearly polarised probe beam propagating through the medium. Generation of the atomic coherences requires the presence of a population imbalance among the Zeeman sublevels. Here, we concentrate on generating a population imbalance representing ‘orientation’, i.e., the majority of a population transferred to the sublevel with the maximum or minimum magnetic number (stretched state), and off-resonance probing [10].

For many practical applications, Size, Weight, Power, and Cost (SWaP-C) considerations are as important as sensor performance. Our study intends to partially address the former without compromising the latter and provides the options to optimise sensor architecture according to the target application. For scenarios where the power consumption sets the main limit, the independent and degenerate pump-probe configuration may be applicable. Improvements to size and cost might be possible with the reduction in the number of laser diodes and relevant electronics, which is achievable in the degenerate and single beam cases. It needs to be stressed that our studies were performed at ambient temperatures in a paraffin-coated cell. In our opinion, miniaturisation of the sensor would have to include wafer-based buffer gas cells, where the required cell heater might become a significant, if not leading, component of the power consumption budget. Activities involving wafer cell development and optimisation are underway in our lab.

A typical orientation generation scheme involves an atomic ensemble’s interaction with a circularly polarised laser beam propagating along the bias magnetic field. Optical pumping also describes the transfer of optical angular momentum to the target atoms via spin-exchange collisions (SEC) [11,12]. Another category of atomic polarisation production combines optical pumping with non-linear (tensor) dynamics driven by linearly polarised light [13,14]. Recently, we have demonstrated a scheme where the orientation within the F = 4 caesium ground-state level is produced with a single linearly polarised laser beam [15,16]. In this case, atomic orientation within the F = 4 level is created by a combination of (1) optical pumping that transfers atoms from the F = 3 into the F = 4 manifold; (2) non-linear dynamics that create an asymmetry in the population distribution within the F = 3 and F = 4 manifolds and (3) SEC that amplify the transfer of the population into a single stretched state by selective relaxation.

In this paper, we compare the performance of three rf atomic magnetometer configurations from the perspective of possible system miniaturisation. The focus is on the efficiency of atomic polarisation generation (measured by spectroscopic profile linewidth, full-width-half-maximum), sensor sensitivity (signal-to-noise ratio), and the laser system parameters required to achieve this sensitivity. In all cases, we study configurations based on indirect pumping [9,17]. The term indirect pumping underlines the lack of direct optical coupling to the target F = 4 level, as depicted in Figure 1a with a pump beam (blue line) tuned to $6{\;}^{2}$S${}_{1/2}$ F = 3 $\to 6{\;}^{2}$P${}_{3/2}$ F’ = 2 and the probe beam (dashed red line) detuned from the $6{\;}^{2}$S${}_{1/2}$ F = 4 $\to 6{\;}^{2}$P${}_{3/2}$ F’ = 5. The first configuration represents a typical case where the pump and probe beams are produced by two independent laser sources, as shown in Figure 1b; the second is an arrangement with two degenerate pump and probe beams that are produced by the same laser, as shown in Figure 1c; and the third is a non-linear scheme with a single linearly polarised beam contributing to orientation generation and off-resonant probing of the system, as shown in Figure 1d, [15]. In all configurations, the pumping process combines optical pumping within the F = 3 manifold, off-resonant population transfer between the F = 3 and F = 4 ground state levels, spin-exchange collisions (SEC), and possible non-linear interactions driven by the probe beam. The role of the SEC is to replicate (spin temperature) the F = 3 population imbalance in the F = 4 level and increase the population of the F = 4 stretched state. Benefits of indirect pumping include a demonstrated level of polarisation (orientation) as high as 92% and reduced power broadening. We use laser power and detuning scans to point out the essential differences between the configurations. We demonstrate through spectroscopic measurements that the different pumping mechanisms produce a comparable level of atomic polarisation within the F = 4 manifold. Consequently, these three configurations offer similar performance with the signal-to-noise ratio limited by atomic noise and quantum back action [18,19].

## 2. Experimental Setup

All measurements described here are performed in a shielded setup. Caesium atomic vapour is housed in a cross-shaped (two intersecting orthogonal cylinders, each with a 35 mm diameter and length) paraffin-coated vapour cell. The vapour cell is enclosed in a 3D-printed oven able to reach temperatures up to 100 °C. We use a waveform generator operating at 1 MHz and an rf amplifier to drive a current in the heater. All of the data reported here were recorded at a temperature of 22 °C (atomic density $n_{\mathrm{Cs}}=3\times 10^{10}$ cm${}^{-3}$). We have performed experiments at an elevated temperature 30 °C ($1.2\times 10^{12}$ cm${}^{-3}$) to confirm trends, e.g., strength of the tensor coupling.

Two separate lasers are used in this work. One can be locked to the hyperfine F = 3 and F = 4 ground-state Caesium transitions via linear absorption spectroscopy. When used to irradiate the atoms, it acts as a pump beam and is tuned in the vicinity of the $6{\;}^{2}$S${}_{1/2}$ F = 3 $\to 6{\;}^{2}$P${}_{3/2}$ F’=2 transition (D2 line, 852 nm), circularly polarised and propagating parallel to the bias static field along $\hat{z}$, as shown in Figure 1b. The second laser is offset locked from the pump beam and can be red/blue detuned by ±10 GHz. This beam is linearly polarised, directed along $\hat{y}$, and can be either used predominately as a probe beam (+1.5 GHz from $6{\;}^{2}$S${}_{1/2}$ F = 4 $\to 6{\;}^{2}$P${}_{3/2}$ F’ = 5), Figure 1b, or as a combined pump and probe beam (0.5–2 GHz from $6{\;}^{2}$S${}_{1/2}$ F = 3 $\to 6{\;}^{2}$P${}_{3/2}$ F’ = 2), as shown in Figure 1c. In the latter configuration, the beam can be split to create a degenerate circularly polarised beam directed along $\hat{z}$ that acts as a typical pump beam. Each beam has a diameter of 10 cm, which is limited by the aperture of the magnetic shields. The beams are expanded to several times this size, giving them a relatively flat intensity power.

Particular care was taken when creating the linearly polarised beam. The polarisation quality was verified with a commercial polarimeter (Thorlabs PAX1000IR1/M). The polarisation for the probe beam light is initially cleaned using a polariser (Glan-Thompson $10^{-3}$ extinction ratio). The beam then propagates through a half-wave and quarter-wave plate before entering the cell. The former sets the linear polarisation angle and the latter compensates for the birefringence introduced by the cell windows.

An rf magnetic field resonant with the Larmor frequency (set by the bias static field) creates coherences between the Zeeman sublevels. This leads to precession of the collective atomic spin generated through optical pumping. Paramagnetic Faraday rotation maps the value of the collective atomic spin onto the polarisation state of the linearly polarised probe beam [8,9,10]. The polarisation rotation is monitored with a simple polarimeter (outputs of a PBS coupled to a balanced photodiode). The resulting signal is demodulated by a lock-in amplifier referenced to the first harmonic of the rf field frequency and outputs the in-phase (*X*) and quadrature (*Y*) component of the signal. The amplitude ($R=\sqrt{X^{2}+Y^{2}}$) of this signal is proportional to the amplitude of the atomic coherence.

The ambient magnetic field is suppressed by the use of Twinleaf cylindrical shields. A pair of solenoids inside the shield generates a well-controlled bias magnetic field with a relative homogeneity at the level of $10^{-4}$ over the length of the cell. A small coil located above the cell aligned along $\hat{x}$ generates the rf magnetic field. The laser beams have been expanded and pass through 10 mm apertures so that only the central part of the beam that has a relatively flat intensity profile irritates the cell. This aperture is limited by the diameter of the optical access in the magnetic shields.

## 3. Comparison between Magnetometer Configurations

Radio-frequency spectroscopy is used to characterise the generation efficiency of the population imbalance (orientation) and the magnetometer signals amplitude. The rf spectrum is created by monitoring the magneto-optical signal as the rf field frequency is scanned across the magnetic resonance. Figure 2 shows a typical rf resonance, recording the *X* (red curve) and *Y* (dark blue curve) components. The resonance frequency, linewidth, and amplitude of the spectral profile have been extracted by fitting a Lorentzian profile to the rf spectrum. We are aware that the Lorentzian profile is produced by a purely orientated state [20,21] and in principle, detailed analysis should include spectral components created by atoms in an aligned and oriented state. Here, we concentrate on a phenomenological description of the signal concerning application to a magnetometer, while a full analysis of the state composition in each configuration will be included elsewhere. The same measurement procedures were used in all magnetometer configurations. We illustrate these methods by discussing the standard magnetometer arrangement involving independent pump and probe lasers.

### 3.1. Independent Pump and Probe

The configuration with independent pump and probe beams is used as a baseline for performance evaluation of the other magnetometer arrangements studied, since it is well understood and has demonstrated high performance [9,17]. In this configuration, the frequency of the pump beam is frequency locked to the $6{\;}^{2}$S${}_{1/2}$ F = 3 $\to 6{\;}^{2}$P${}_{3/2}$ F’ = 2 transition, while the frequency of the probe beam is locked 1.5 GHz from this transition.

Optimisation of the rf atomic magnetometer signal begins with a selection of test rf field amplitude values that will be used in all further experiments. The rf field amplitude is set such that it does not perturb the linewidth of the spectroscopic measurements. The value is selected by recording the dependence of the rf spectroscopy signal on the rf field amplitude. This dependence is also used in rf field calibration [9].

Figure 3 shows the amplitude (blue squares) and linewidth (red dots) of the rf spectroscopy profile as a function of the pump beam power. We interpret the narrowing visible in Figure 3b as a result of a reduction of SEC relaxation due to increased polarisation within the atomic vapour. Subsequent line broadening observed for high pump powers is caused by the dominance of pumping to the stretched state, causing coherence oscillation relaxation.

The dependence of the rf atomic magnetometer signal on the probe beam power, as shown by the blue squares in Figure 4a, was recorded using the pump beam power that maximises signal amplitude and minimises the rf profile linewidth. The dependence is linear over the majority of the probe beam power range. As discussed in ref. [16], the range where a linear power dependence is observed is related to how far detuned the probe beam is from the F = 4 transitions; e.g., saturation occurs at higher powers for larger detunings. Figure 4b compares the signal linewidth dependencies on the probe beam power recorded at two probe laser detunings, 0.972 GHz (red dots) and 0.486 GHz (green diamonds). The data sets indicate the presence of a mechanism driven by the probe laser that leads to the narrowing of the rf resonance profiles. The linearly polarised light (${\vec{E}}_{p}$) couples to the atomic ground state through the tensor ac polarizability $\alpha_{2}$ (single-spin Hamiltonian, without the scalar part of the light shift $∼\alpha_{2}{\left({\vec{E}}_{p}\;\cdot \;{\hat{f}}^{\left(i\right)}\right)}^{2}$, where ${\hat{f}}^{\left(i\right)}$ is the total angular momentum operator of the i’th atom). Therefore, in general, the atomic spin dynamics will exhibit a non-linear character at high probe laser powers [13,14]. The effect, which is enhanced at elevated temperatures, is equivalent to a torque resulting from the combination of the bias magnetic and linearly polarised probe beam [14].

The phenomenon is similar to the well-known alignment-to-orientation conversion [22,23,24] and leads to an increase in atomic polarisation (orientation) and narrowing of the spectral profile. The narrowing is accompanied by a departure from the linear increase in amplitude. The influence this narrowing has on the magnetometer’s performance will be discussed in a future paper. Here, it is only worth pointing out that the narrowing of the rf profile and the accompanying increase of its amplitude does not translate directly to an increase in the signal-to-noise ratio.

To evaluate the signal-to-noise ratio, noise spectra, such as presented in Figure 4c, were recorded with a spectrum analyser (1Hz bandwidth) at different probe beam powers. Analysis of the noise spectrum and its probe power dependence, shown as the black points in Figure 4a, allows the identification of various noise components. Electronic noise, indicated by a black line in Figure 4c, does not show dependence on either the probe power or rf frequency within the relevant range [flat part in Figure 4a]. Photonic shot noise, the green line in Figure 4c, can be recognised as the flat background. Its amplitude increases with probe beam power. The major noise component comes from the atomic projection noise and quantum back-action (recorded without an rf field). As a consequence of the fluctuation–dissipation theorem, the spectral profile of the atomic noise mirrors the profile observed in the rf spectrum. It is important to point out that these studies are performed at a relatively low operating frequency, 5 kHz and benefit from composite shielding, particularly the ferrite inner layer of the shields. Measurements at a similar frequency performed in a different system with 5 mu-metal shields are dominated by Johnson noise (at 10 fT Hz${}^{-1/2}$ level) produced by the inner layer of mu-metal.

The probe beam power amplitude dependencies of the signal and noise were used to evaluate the signal-to-noise ratio. The ratios for the three discussed configurations are presented in Figure 5.

### 3.2. Degenerate Pump and Probe

The degenerate configuration refers to a case where the pump and probe beams are produced by a single laser source, i.e., have the same frequency, but different polarisations, as shown in Figure 1c. In contrast to the previous case, the pump and probe beams are detuned significantly away from the relevant atomic resonant transitions, e.g., 0.5 GHz from the $6{\;}^{2}$S${}_{1/2}$ F = 3 $\to 6{\;}^{2}$P${}_{3/2}$ F’ = 2 transition.

Previously, we have shown that changing the probe beam detuning does not affect the maximum achievable signal-to-noise ratio [16]. With increasing probe laser detuning, the power for which the maximum value is achieved increases. However, in the degenerate and single beam cases, the probe has the same frequency as the pump beam, and optimisation of the laser detuning becomes an inherent part of the magnetometer performance optimisation process. To demonstrate the role of laser detuning in a degenerate pump and probe configuration, we have recorded the laser detuning dependency of the rf spectra. The measurement was calculated by tuning a voltage-controlled oscillator that sets the detuning offset frequency of the laser relative to a second beam that is frequency locked to the atomic transition. Figure 6 shows the typical dependence of the rf spectra on laser detuning. The plot confirms that the magnetometer signal, i.e., the F = 4 coherence, can be observed in a wide frequency range on the low-frequency side of the $6{\;}^{2}$S${}_{1/2}$ F = 3 $\to 6{\;}^{2}$P${}_{3/2}$ F’ = 2 transition. The optimum of the signal can be observed over a 100 MHz frequency range. For detunings >0.7 GHz, the efficiency of pumping atoms into the F = 4 manifold decreases and a signature of the F = 3 atoms becomes visible. The position of the F = 3 resonance changes with laser detuning, which is caused by a Zeeman light shift introduced by the pump beam.

Figure 7 shows the amplitude (a) and linewidth (b) of the rf spectral profiles as a function of laser detuning recorded for three pump beam powers: 4.83 mW (blue dots), 1.169 mW (green squares), and 0.067 mW (red triangles). In these measurements, the probe beam power was kept constant at 2.33 mW level. The probe beam is 9 GHz from transitions relevant to the F = 4 ground state, so the change in detuning predominately affects the pumping process. The plots confirm the presence of a power that optimises pumping, in terms of amplitude and linewidth, and that the range of the frequencies where the optimum signal is observed moves away from the resonant transition with increasing pump power. It is worth stressing that pumping is optimised at a non-zero detuning. We interpret the presence of the optimum pumping at the non-zero detuning in terms of the balance between the direct pumping (orientation generation) within the F = 3 level and the transfer of the population to the F = 4 level.

To address the issue of optimum pump beam power more specifically, we measured the rf spectral profile parameters as a function of the pump beam power for fixed probe beam power and detuning in the vicinity of the optimum detuning from Figure 6. The pump power dependence of the signal amplitude and linewidth presented in Figure 8 shows that the character of pumping through the off-resonant pump beam results in a level of atomic polarisation similar to that observed in the independent pump and probe configuration. The range of pump powers that optimises the magnetometer signal is narrower, i.e., between 0.7 mW and 1.5 mW, than in the case where two independent beams are used. This is due to pumping with a beam whose frequency is tuned away from atomic resonance. Comparisons performed at 22 °C and 30 °C confirm that the smallest rf profile linewidth observed in the degenerate configuration is within 10% of that observed in independent pump and probe arrangement.

Detuning scans, such as Figure 6, recorded at a constant pump power show an increase in the optimum detuning for increasing probe powers without compromising orientation generation, i.e., the same linewidth. This indicates that the role of the linearly polarised probe beam is not restricted solely to the optical readout of the atomic signal. This could be explained in terms of the probe-driven non-linear contribution to the orientation generation within the F = 4 manifold. While a detailed discussion of this contribution goes beyond the scope of this paper, it is worth pointing out that in contrast to the case of independent pump and probe, the non-linear coupling acts on the F = 3 atoms, and its results are replicated in the F = 4 atoms through SEC and off-resonant pumping. To further verify this, we performed a test comparing the rf spectrum recorded using a strong probe beam (20 mW) and two orthogonal circular ($\sigma_{+}$ and $\sigma_{-}$) pump polarisations, similar to the data illustrated Figure 4b in ref. [15]). We saw a change by a factor of five in signal amplitude for these conditions, confirming the probe beam is contributing to the pumping of a particular stretched state. In other words, the linearly polarised beam produces an orientation that either compeates or assists with the orientation generated by the pump beam.

Figure 9 shows the amplitude (blue squares) and linewidth (red dots) of the rf spectroscopy profile as a function of the probe beam power. Noise spectra were recorded for these powers and used to determine the SNR for this configuration as shown in Figure 5. The results are discussed after the introduction of the final laser configuration.

### 3.3. Single Beam Configuration

The single beam configuration refers to a case where orientation within the F = 4 caesium ground state level is produced and probed by a single linearly polarised laser beam, as shown in Figure 1d. Non-linear interactions driven by the high-power linearly polarised laser beam and spin-exchange collisions between the two ground states define the atomic polarisation (orientation) within the F = 4 level. This differs from the polarisation typically associated with a linearly polarised pump beam (alignment), as demonstrated in the Lorentzian rf spectral profile and the dependence of its amplitude on the laser beam power.

Figure 10 shows the dependence of the rf signal amplitude and linewidth on the laser beam power. For beam powers below 2 mW, spectra have the form typical for alignment [15]. This results from the population imbalance created directly by linearly polarised light within the F = 3 ground state manifold. This population imbalance (alignment) is mirrored in the F = 4 level through off-resonant pumping. The spectral character changes for the laser beam powers around 2 mW. A combination of high-power linearly polarised light and the bias magnetic field provides a torque to the atoms in the same way as discussed previously, changing the character of the population imbalance character that leads to orientation polarisation. Data in the plot represent the parameters of the Lorentzian fit to the rf spectral profiles representing orientation. The non-linear increase of the signal amplitude with the laser beam power reflects the non-linear character of the underlying mechanism. The linewidth power dependence, i.e., narrowing, is similar to that observed with independent pump and probe lasers, as shown in Figure 3. However, the minimum linewidth in Figure 10 is larger than recorded in the independent pump/ probe configuration. This is due to a decreased pumping efficiency, which is relative to when using a circularly polarised beam directed along the bias field.

Rf spectra were systematically recorded for various laser detunings from the $6{\;}^{2}$S${}_{1/2}$ F = 3 $\to 6{\;}^{2}$P${}_{3/2}$ F’ = 2 transition, as shown in Figure 11. The recording was performed with a laser power of 12.56 mW where orientation could be observed around 0.5 GHz. This plot was recorded for a range of beam powers, and the optimum detuning was seen to increase with beam power, similarly to that seen in Figure 7. The relatively narrow resonant character of the signal amplitude dependence on detuning could be used for frequency stabilisation.

Similarly to other cases, the noise spectra were recorded with a spectrum analyser for the laser beam power values used in the previous measurement. In agreement with the fluctuation–dissipation theorem, the changes in the character of the rf spectroscopy profiles were also observed in the atomic noise profiles. In particular, the noise spectra recorded below 7 mW contain contributions from the noise produced by alignment within the F = 3 ground-state level.

### 3.4. Comparison between Magnetometer Configurations

The three magnetometer configurations presented here represent three different indirect pumping mechanisms to generate atomic orientation. The first two involve circularly polarised pump beams, while the third is solely based on a single linearly polarised beam. While the efficiency of pumping in terms of the rf resonance linewidth is comparable, the mechanisms differ and are highlighted by the conditions required to achieve optimum performance. The independent pump and probe configuration and degenerate arrangement differ in terms of pumping not only in the power of the pump beam that is required to optimise the process but also in the range of powers for which the optimum is observed. This is a direct consequence of the difference in pump beam detuning from the $6{\;}^{2}$S${}_{1/2}$ F = 3 $\to 6{\;}^{2}$P${}_{3/2}$ F’ = 2 transition. Pumping with the single beam configuration is optimised at even higher powers and over a narrower range.

While discrimination between different magnetometer configurations through direct comparison of the signal amplitudes is meaningless because of the different probe beam detunings and powers, the signal-to-noise ratio provides a tool for the evaluation of different arrangements’ performance. Figure 5 shows the dependence of the signal-to-noise ratio on the linear polarised beam power measured in the three configurations. For the two beam configurations involving a circularly polarised pump beam, this is a power scan of the probe beam power. The data for these two configurations show a linear power dependence before saturation. The data set for the single beam configuration shows a quadratic dependence that reflects the difference in the underlying pumping mechanisms. This difference is also reflected in the beam power for which the optimum signal-to-noise ratio is achieved. It can be seen that the optimum signal-to-noise ratio is the same for all configurations.

The standard configuration with independent pump and probe beams offers the optimum signal-to-noise ratio at the lowest power, which is easily accessible with a standard vertical cavity surface emitting laser (VCSEL) diode. This comes at the price of the relative complexity of the instrumentation: two separate diodes, drivers and stabilisation. Frequency locking of the pump beam is achievable with standard absorption spectroscopy, but securing stable off-resonant detuning (GHz range) of the probe beam is more complex. A method used in atomic clocks [25] involves generating optical laser frequency sidebands by modulating the laser drive current. Either component can then be locked to an atomic resonance transition (pump), and the other can be used for probing. The laser beam power required for the degenerate pump and probe beam configuration is also within the range of a VCSEL diode [26], and the resonance feature presented in Figure 6 could be used for the laser frequency locking. VCSEL diodes are currently unavailable with the output power required to optimise the performance of the single beam configuration, but an increase of the diode diaphragm and doping of the active region could make a high power source available.

## 4. Conclusions

To summarise, we have systematically explored different laser configurations of an rf atomic magnetometer, based on similar pumping mechanisms, with variation in the role the probe beam plays. We showed the generation of a similar level of orientation in all three cases through various types of pumping mechanisms. Pumping involves a circularly polarised beam for the independent pump and probe configuration, both circularly and linearly polarised beams in the degenerate case, and solely a linearly polarised beam when there is a single beam. We demonstrate the same signal-to-noise ratio in all three arrangements. As mentioned, the presence of non-linear interactions driven by the linearly polarised beam creates options for improved orientation within the F = 4 manifold, operating an all-optical internally driven non-linear spin maser [14] and spin squeezing [24]. The influence of these effects on magnetometer performance will be addressed in our following works.

## Figures and Tables

**Figure 1 sensors-22-09741-f001:**
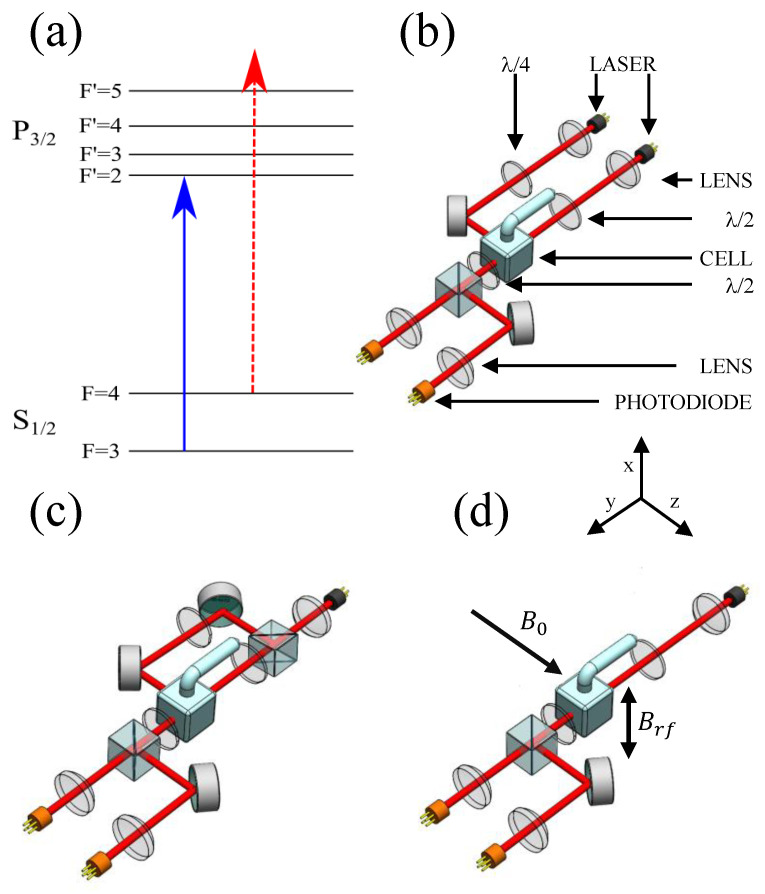
(**a**) Energy structure of cesium $6{\;}^{2}$S${}_{1/2}$$\to 6{\;}^{2}$P${}_{3/2}$ transition (D2 line, 852 nm). The solid blue arrow represents the frequency of the pump beam, while the dashed red arrow represent the relevant frequency of the probe laser used in the independent pump and probe configuration. There are three possible configurations of the radio-frequency atomic magnetometer: (**b**) A circularly-polarised laser light (pump), tuned to the F = 3→F’ = 2 transition propagating along $\hat{z}$, transfers atoms into the stretched state of the F=4 ground state level. The probe beam, propagating along $\hat{y}$, monitors the spin precession via the paramagnetic Faraday effect. Both beams are at the same frequency, blue detuned 1GHz from the F = 4→F’ = 5 transition (D2 line). (**c**) A single beam is split into a circularly polarised beam propagating along the $\hat{z}$ axis, and a linearly polarised beam propagating along $\hat{y}$ is used to create the population imbalance within the F = 4 caesium ground state manifolds. The linearly polarised beam acts also as a probe. (**d**) A linearly polarised laser beam, acting as the pump, transfers the population between and creates a population imbalance within the F = 4 caesium ground state manifolds. The same linearly polarised laser beam acts as the probe.

**Figure 2 sensors-22-09741-f002:**
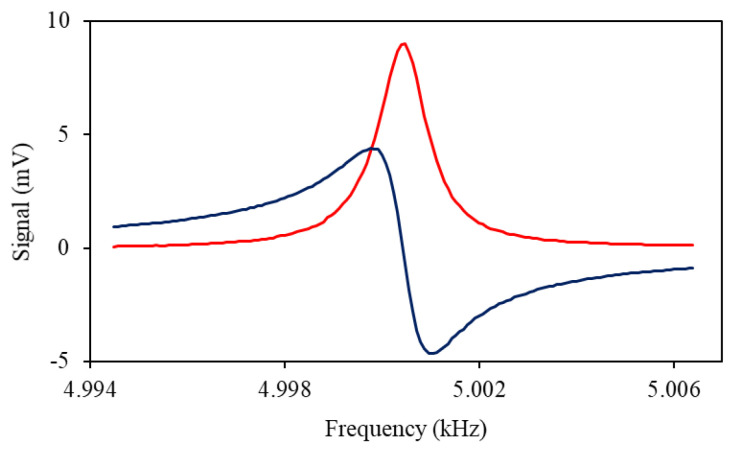
Recorded rf spectrum, consisting of the two quadrature lock-in amplifier outputs.

**Figure 3 sensors-22-09741-f003:**
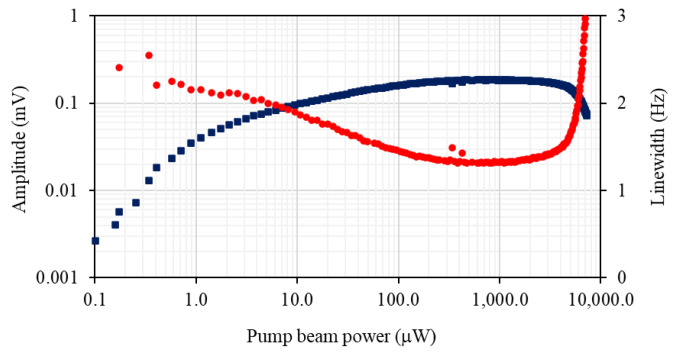
Amplitude (blue squares) and linewidth (red dots) of the rf spectral profile as a function of pump beam power recorded at cell temperature of 22 °C.

**Figure 4 sensors-22-09741-f004:**
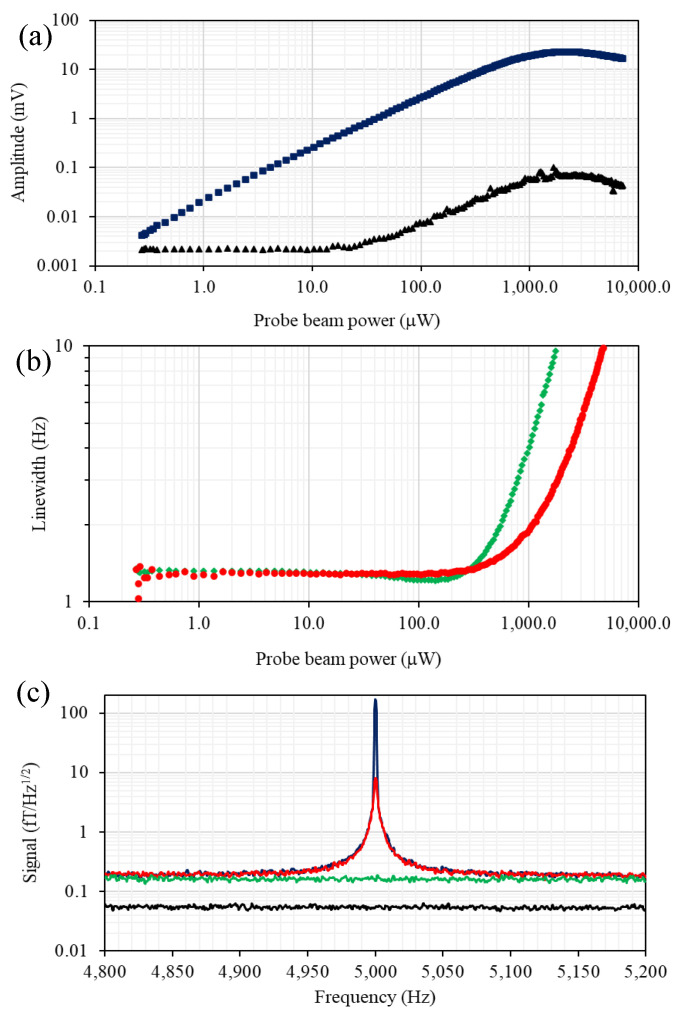
(**a**) The amplitude and linewidth (red dots) of the rf spectral profile as a function of pump beam power recorded at cell temperature of 22 °C. of the magnetometer signal versus the probe-beam power (blue squares). The sum of the various noise components (electronic noise, photonic-shot noise, atomic-projection noise and probe-beam back-action) is depicted by the black triangles. (**b**) The linewidth of the magnetometer signal as a function of probe beam power recorded at two probe laser detunings, 0.972 GHz (red dots) and 0.486 GHz (green diamonds). (**c**) Frequency spectrum of the radio-frequency magnetometer recorded with (blue solid line) and without (red dashed line) calibrated rf magnetic field. The photonic-shot-noise level represented by the green line and the black line describes the baseline electronic noise. The resolution bandwidth for the measurement was 1 Hz, pump beam power was 750 μW, probe beam power was 200 μW, and the atomic density was 3.3 × 10^10^ cm^−1^.

**Figure 5 sensors-22-09741-f005:**
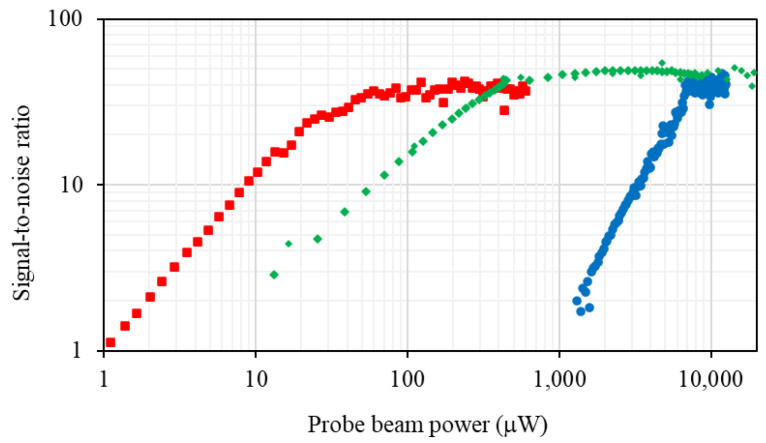
Signal-to-noise ratio as a function of the probe beam power for three different rf atomic magnetometer configurations: two independent pump and probe beams (red squares), a degenerate pump and probe (green diamonds), and a single beam (blue circles).

**Figure 6 sensors-22-09741-f006:**
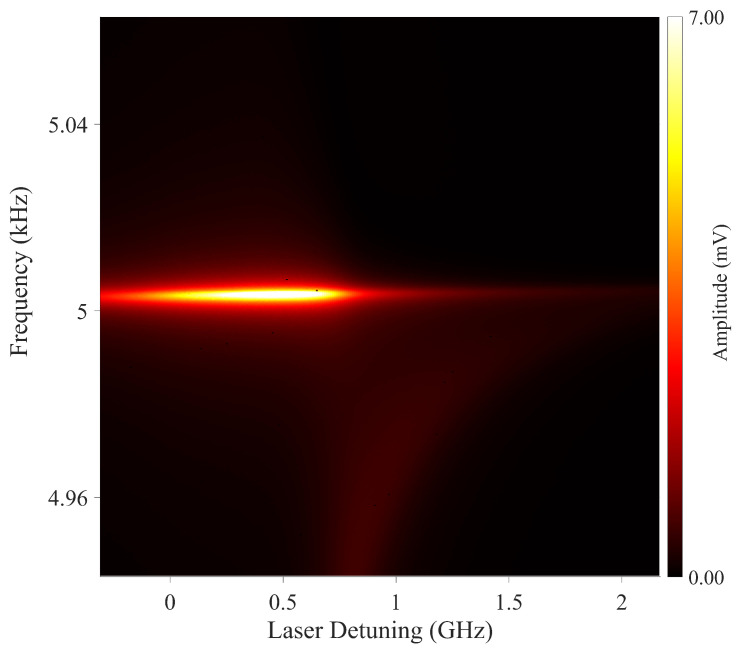
Dependence of the rf signal amplitude on the probe beam detuning from the $6{\;}^{2}$S${}_{1/2}$ F = 3$\to 6{\;}^{2}$P${}_{3/2}$ F’ = 2 transition. The measurements were made with a pump beam power of 1.17 mW and probe beam power of 1 mW.

**Figure 7 sensors-22-09741-f007:**
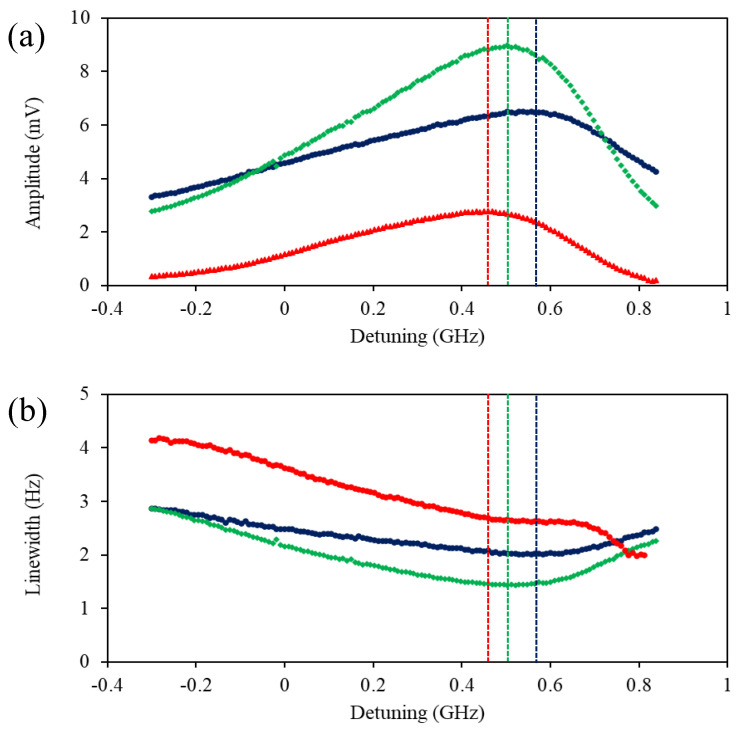
The amplitude (**a**) and the linewidth (**b**) of the rf spectrum profile as a function of laser detuning in the degenerate pump and probe beam configuration recorded for the pump beam power of 4.83 mW (blue dots), 1.169 mW (green squares), and 0.067 mW (red triangles).

**Figure 8 sensors-22-09741-f008:**
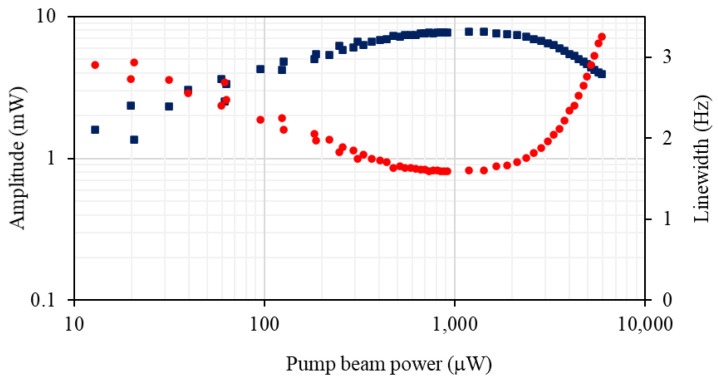
Amplitude (blue squares) and linewidth (red dots) of the rf spectral profile as a function of pump beam power for the degenerate case.

**Figure 9 sensors-22-09741-f009:**
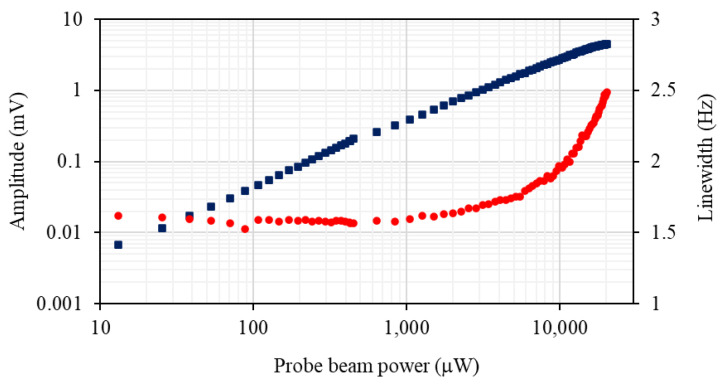
Amplitude (blue squares) and linewidth (red dots) of the rf spectral profile as a function of probe beam power.

**Figure 10 sensors-22-09741-f010:**
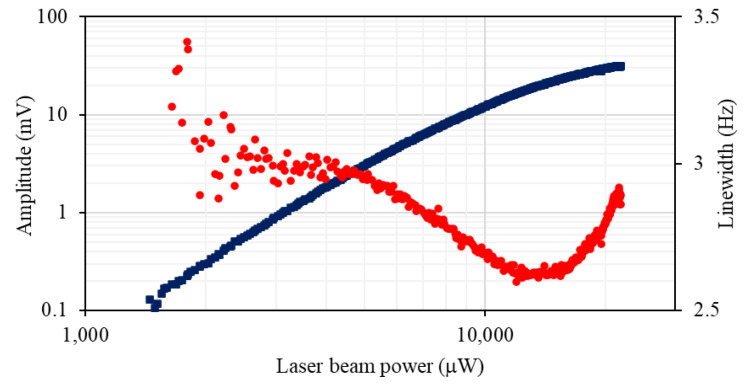
The amplitude (blue squares) and linewidth (red dots) of the rf spectrum profile versus the laser beam power in a single beam configuration.

**Figure 11 sensors-22-09741-f011:**
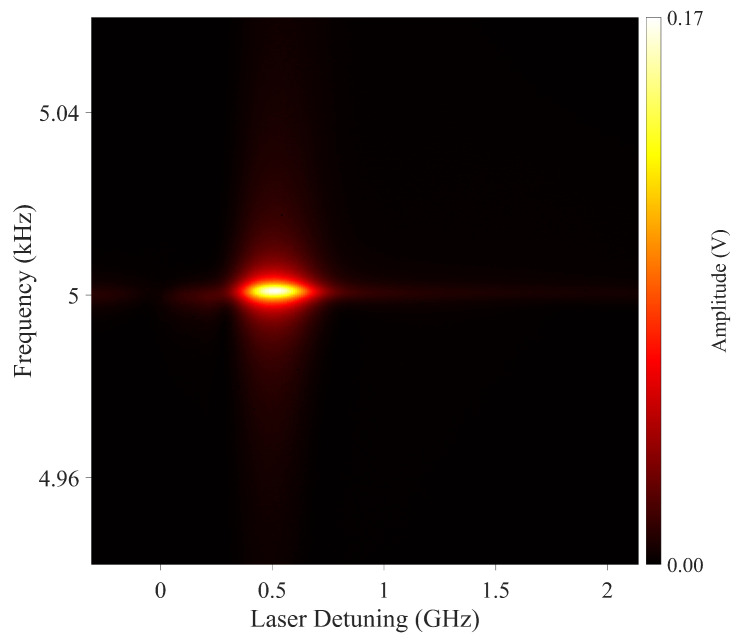
Dependence of the rf signal amplitude on the linearly polarised beam detuning from the $6{\;}^{2}$S${}_{1/2}$ F = 3$\to 6{\;}^{2}$P${}_{3/2}$ F’ = 2 transition. The measurements were made with a laser beam power of 12.56 mW.

## Data Availability

The data that support the findings of this study are available from the corresponding author upon reasonable request.

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
