# Peer review of "Different Configurations of Radio-Frequency Atomic Magnetometers—A Comparative Study"

_sensors, 2022, doi:10.3390/s22249741_

Round 1

Reviewer 1 Report

In the manuscript entitled “Different configurations of radio-frequency atomic magnetometers - a comparative study” that is submitted the Journal Sensors, the authors compared the power and detection performances for the atomic magnetometers with three configurations: (a) the typical pump/probe with two independent sources (b) two degenerated pump and probe beams from a same laser (c) a linearly polarized laser beam that acts as the pump and the probe. As such, the size of the atomic magnetometer can be reduced. It is also concluded that the three configurations can result in the same level of the signal-to-noise ratio. However, the instruments are simplified at a cost of the power consumption in this case.

I recommend the author to emphasize the importance of miniaturization. Besides, the power consumption of ~10mW seems a little bit high for some applications, such as the long-term observation of the environmental magnetic fields.

How to choose the optimal vapor pressure for the sensor head? Is it based on the calculations or on the experiments?

Does the robustness is analyzed for the configuration with a linearly polarized laser beam?

To further reduce the power consumption, can the vapour density be lowered?

Author Response

Q1) How to choose the optimal vapor pressure for the sensor head? Is it based on the calculations or on the experiments?

            The cell used in these studies is a paraffin-coated cell, and all of the measurements reported in the manuscript were performed at room temperature. We conducted our studies at ambient temperature to avoid complications due to non-linear effects that were observed at elevated temperatures (without affecting SNR).

In parallel, we made the same optimization measurements for all configurations at ~30degC. The signal-to-noise ratio was higher by a factor of 2, which is in agreement with our previous observations, ref (W. Chalupczak, R. M. Godun, S. Pustelny, W. Gawlik, Room temperature femtotesla radio-frequency atomic magnetometer, Appl. Phys. Lett., 100, 242401, 2012).

Q2) Does the robustness is analyzed for the configuration with a linearly polarized laser beam?

We have concentrated on the physics of operation in different configurations, without exploring the technical details of particular options. While the single beam configuration ensures simplicity of instrumentation, we recognise that it takes advantage of non-linear processes that depend on decoherence rate. The decoherence rate is significantly higher in miniature buffer gas cells and we are currently working on verifying the limits of coherence lifetime in silicon wafer-based cell.

Q3) I recommend the author to emphasize the importance of miniaturization. Besides, the power consumption of ~10mW seems a little bit high for some applications, such as the long-term observation of the environmental magnetic fields.

To further reduce the power consumption, can the vapour density be lowered?

            The paraffin-coated vapour cell is operated at room temperature and our previous studies indicated that optimum is achieved at ~30degC.

In future stages of development towards commercialization, a miniature cell with buffer gas will be implemented. To ensure proper signal heating is required and heating becomes significant in terms of power consumption. We have included a comment regarding SWaP-C considerations in the text. We added the text below to the bottom of page 3.

‘For many practical applications, Size, Weight, Power, and Cost (SWaP-C) considerations are as important as sensor performance. Our study intends to address partially address the former without compromising the latter and provides the options to optimise sensor architecture according to the target application. For scenarios where the power consumption set the main limit, the independent and degenerate pump-probe configuration may be applicable. Improvements to size and cost might be possible with the reduction in the number of laser diodes and relevant electronics, which is achievable in the degenerate and single beam cases. It needs to be stressed that our studies were performed at ambient temperatures in a paraffin-coated cell. In our opinion, miniaturisation of the sensor would have to include wafer-based buffer gas cells, where the required cell heater might become a significant, if not leading, component of the power consumption budget.  Activities involving wafer cell development and optimisation are underway in our lab.’

Reviewer 2 Report

The article is on a very good level.

I have only minor comments:

Missing reference to Figure 1 in the text In the description of Figure 1, consider whether to list polarised as polarized (similarly below) and caesium as cesium and represent as represents.

For Figure 3, put the description closer to the figure so that the gap is no more than between the description and the following text.

Similarly, edit for most other figures.

Line 209 consider changing optimises to optimizes, similarly in other places.

Figure 11 – It would be appropriate to put in front of Conclusions.

Author Response

Q1) Missing reference to Figure 1 in the text In the description of Figure 1,

Setups (b,c) are described in the paragraph at the top of page 3 and again in the relevant sections of the manuscript. Figure 1 (d) is referred to in its corresponding section. We have added a section to the top of page 3 to describe Fig.1 (a), that text is copied below.

‘The term indirect pumping underlines the lack of direct optical coupling to the target  F=4 level, as depicted in Fig.~\ref{fig:Fig1_Setup} (a) with pump beam (blue line) tuned to $6\,^2$S$_{1/2}$  F = 3 $\rightarrow{} 6\,^2$P$_{3/2}$ F`=2 and the probe beam (dashed red line) detuned from the $6\,^2$S$_{1/2}$  F = 4 $\rightarrow{} 6\,^2$P$_{3/2}$ F`=5.’

Q2) consider whether to list polarised as polarized (similarly below) and caesium as cesium and represent as represents.

We will defer to the editor’s advice regarding language changes

Q3) For Figure 3, put the description closer to the figure so that the gap is no more than between the description and the following text. Similarly, edit for most other figures.

White spacing was removed from this figure and figures 2, 7-9, and 11.

Q4) Line 209 consider changing optimises to optimizes, similarly in other places.

            See Q2

Q5) Figure 11 – It would be appropriate to put in front of Conclusions.

            Figure placement has been adjusted.